# WEIGHTED TRANSFORMER NETWORK FOR MACHINE TRANSLATION

## ABSTRACT

State-of-the-art results on neural machine translation often use attentional sequence-to-sequence models with some form of convolution or recursion. Vaswani et al. (2017) propose a new architecture that avoids recurrence and convolution completely. Instead, it uses only self-attention and feed-forward layers. While the proposed architecture achieves state-of-the-art results on several machine translation tasks, it requires a large number of parameters and training iterations to converge. We propose Weighted Transformer, a Transformer with modified attention layers, that not only outperforms the baseline network in BLEU score but also converges $15 - 40\%$ faster. Specifically, we replace the multi-head attention by multiple self-attention branches that the model learns to combine during the training process. Our model improves the state-of-the-art performance by $0.5$ BLEU points on the WMT 2014 English-to-German translation task and by $0.4$ on the English-to-French translation task.

## 1 INTRODUCTION

Recurrent neural networks (RNNs), such as long short-term memory networks (LSTMs) (Hochreiter & Schmidhuber, 1997), form an important building block for many tasks that require modeling of sequential data. RNNs have been successfully employed for several such tasks including language modeling (Melis et al., 2017; Merity et al., 2017; Mikolov), speech recognition (Xiong et al., 2017; Graves et al., 2013; Lee et al., 1995), and machine translation (Wu et al., 2016; Bahdanau et al., 2014). RNNs make output predictions at each time step by computing a hidden state vector $h_t$ based on the current input token and the previous states. This sequential computation underlies their ability to map arbitrary input-output sequence pairs. However, because of their auto-regressive property of requiring previous hidden states to be computed before the current time step, they cannot benefit from parallelization.

Variants of recurrent networks that use strided convolutions eschew the traditional time-step based computation (Kaiser & Bengio, 2016; Lei & Zhang, 2017; Bradbury et al., 2016; Gehring et al., 2016; 2017; Kalchbrenner et al., 2016). However, in these models, the operations needed to learn dependencies between distant positions can be difficult to learn (Hochreiter et al., 2001; Hochreiter, 1998). Attention mechanisms, often used in conjunction with recurrent models, have become an integral part of complex sequential tasks because they facilitate learning of such dependencies (Luong et al., 2015; Bahdanau et al., 2014; Parikh et al., 2016; Paulus et al., 2017; Kim et al., 2017).

In Vaswani et al. (2017), the authors introduce the Transformer network, a novel architecture that avoids the recurrence equation and maps the input sequences into hidden states solely using attention. Specifically, the authors use positional encodings in conjunction with a multi-head attention mechanism. This allows for increased parallel computation and reduces time to convergence. The authors report results for neural machine translation that show the Transformer networks achieves state-of-the-art performance on the WMT 2014 English-to-German and English-to-French tasks while being orders-of-magnitude faster than prior approaches.

Transformer networks still require a large number of parameters to achieve state-of-the-art performance. In the case of the newstest2013 English-to-German translation task, the base model required 65M parameters, and the large model required 213M parameters. We propose a variant of the Transformer network which we call Weighted Transformer that uses self-attention branches in lieu of the multi-head attention. The branches replace the multiple heads in the attention mechanism of

the original Transformer network, and the model learns to combine these branches during training. This branched architecture enables the network to achieve comparable performance at a significantly lower computational cost. Indeed, through this modification, we improve the state-of-the-art performance by 0.5 and 0.4 BLEU scores on the WMT 2014 English-to-German and English-to-French tasks, respectively. Finally, we present evidence that suggests a regularizing effect of the proposed architecture.

## 2 RELATED WORK

Most architectures for neural machine translation (NMT) use an encoder and a decoder that rely on deep recurrent neural networks like the LSTM (Luong et al., 2015; Sutskever et al., 2014; Bahdanau et al., 2014; Wu et al., 2016; Barone et al., 2017; Cho et al., 2014). Several architectures have been proposed to reduce the computational load associated with recurrence-based computation (Gehring et al., 2016; 2017; Kaiser & Bengio, 2016; Kalchbrenner et al., 2016). Self-attention, which relies on dot-products between elements of the input sequence to compute a weighted sum (Lin et al., 2017; Bahdanau et al., 2014; Parikh et al., 2016; Kim et al., 2017), has also been a critical ingredient in modern NMT architectures. The Transformer network (Vaswani et al., 2017) avoids the recurrence completely and uses only self-attention.

We propose a modified Transformer network wherein the multi-head attention layer is replaced by a branched self-attention layer. The contributions of the various branches is learned as part of the training procedure. The idea of multi-branch networks has been explored in several domains (Ahmed & Torresani, 2017; Gastaldi, 2017; Shazeer et al., 2017; Xie et al., 2016). To the best of our knowledge, this is the first model using a branched structure in the Transformer network. In Shazeer et al. (2017), the authors use a large network, with billions of weights, in conjunction with a sparse expert model to achieve competitive performance. Ahmed & Torresani (2017) analyze learned branching, through gates, in the context of computer vision while in Gastaldi (2017), the author analyzes a two-branch model with randomly sampled weights in the context of image classification.

### 2.1 TRANSFORMER NETWORK

The original Transformer network uses an encoder-decoder architecture with each layer consisting of a novel attention mechanism, which the authors call multi-head attention, followed by a feed-forward network. We describe both these components below.

From the source tokens, learned embeddings of dimension $d_{\text{model}}$ are generated which are then modified by an additive positional encoding. The positional encoding is necessary since the network does not otherwise possess any means of leveraging the order of the sequence since it contains no recurrence or convolution. The authors use additive encoding which is defined as:

$$\text{PE}(pos, 2i) = \sin(pos/10000^{2i/d_{\text{model}}})$$
$$\text{PE}(pos, 2i + 1) = \cos(pos/10000^{2i/d_{\text{model}}}),$$

where $pos$ is the position of a word in the sentence and $i$ is the dimension of the vector. The authors also experiment with learned embeddings (Gehring et al., 2016; 2017) but found no benefit in doing so. The encoded word embeddings are then used as input to the encoder which consists of $N$ layers each containing two sub-layers: (a) a multi-head attention mechanism, and (b) a feed-forward network.

A multi-head attention mechanism builds upon scaled dot-product attention, which operates on a query $Q$, key $K$ and a value $V$:

$$\text{Attention}(Q, K, V) = \text{softmax}\left(\frac{QK^T}{\sqrt{d_k}}\right) V \tag{1}$$

where $d_k$ is the dimension of the key.

In the first layer, the inputs are concatenated such that each of $(Q, K, V)$ is equal to the word vector matrix. This is identical to dot-product attention except for the scaling factor $d_k$, which improves numerical stability.

Multi-head attention mechanisms obtain $h$ different representations of $(Q, K, V)$, compute scaled dot-product attention for each representation, concatenate the results, and project the concatenation with a feed-forward layer. This can be expressed in the same notation as Equation (1):

$$\text{head}_i = \text{Attention}(QW_i^Q, KW_i^K, VW_i^V) \tag{2}$$

$$\text{MultiHead}(Q, K, V) = \text{Concat}_i(\text{head}_i)W^O \tag{3}$$

where the $W_i$ and $W^O$ are parameter projection matrices that are learned. Note that $W_i^Q \in \mathbb{R}^{d_{\text{model}} \times d_k}$, $W_i^K \in \mathbb{R}^{d_{\text{model}} \times d_k}$, $W_i^V \in \mathbb{R}^{d_{\text{model}} \times d_v}$ and $W^O \in \mathbb{R}^{h d_v \times d_{\text{model}}}$ where $h$ denotes the number of heads in the multi-head attention. Vaswani et al. (2017) proportionally reduce $d_k = d_v = d_{\text{model}}/h$ so that the computational load of the multi-head attention is the same as simple self-attention.

The second component of each layer of the Transformer network is a feed-forward network. The authors propose using a two-layered network with a ReLU activation. Given trainable weights $W_1, W_2, b_1, b_2$, the sub-layer is defined as:

$$\text{FFN}(x) = \max(0, xW_1 + b_1)W_2 + b_2 \tag{4}$$

The dimension of the inner layer is $d_{ff}$ which is set to $2048$ in their experiments. For the sake of brevity, we refer the reader to Vaswani et al. (2017) for additional details regarding the architecture.

For regularization and ease of training, the network uses layer normalization (Ba et al., 2016) after each sub-layer and a residual connection around each full layer (He et al., 2016). Analogously, each layer of the decoder contains the two sub-layers mentioned above as well as an additional multi-head attention sub-layer that receives as inputs $(V, K)$ from the output of the corresponding encoding layer. In the case of the decoder multi-head attention sub-layers, the scaled dot-product attention is masked to prevent future positions from being attended to, or in other words, to prevent illegal leftward-ward information flow.

One natural question regarding the Transformer network is why self-attention should be preferred to recurrent or convolutional models. Vaswani et al. (2017) state three reasons for the preference: (a) computational complexity of each layer, (b) concurrency, and (c) path length between long-range dependencies. Assuming a sequence length of $n$ and vector dimension $d$, the complexity of each layer is $\mathcal{O}(n^2 d)$ for self-attention layers while it is $\mathcal{O}(nd^2)$ for recurrent layers. Given that typically $d > n$, the complexity of self-attention layers is lower than that of recurrent layers. Further, the number of sequential computations is $\mathcal{O}(1)$ for self-attention layers and $\mathcal{O}(n)$ for recurrent layers. This helps improved utilization of parallel computing architectures. Finally, the maximum path length between dependencies is $\mathcal{O}(1)$ for the self-attention layer while it is $\mathcal{O}(n)$ for the recurrent layer. This difference is instrumental in impeding recurrent models' ability to learn long-range dependencies.

## 3 PROPOSED NETWORK ARCHITECTURE

We now describe the proposed architecture, the Weighted Transformer, which is more efficient to train and makes better use of representational power.

In Equations (3) and (4), we described the attention layer proposed in Vaswani et al. (2017) comprising the multi-head attention sub-layer and a FFN sub-layer. For the Weighted Transformer, we propose a branched attention that modifies the entire attention layer in the Transformer network (including both the multi-head attention and the feed-forward network).

The proposed attention layer can be mathematically described as:

$$\text{head}_i = \text{Attention}(QW_i^Q, KW_i^K, VW_i^V), \tag{5}$$

$$\overline{\text{head}}_i = \text{head}_i W^{O_i} \times \kappa_i, \tag{6}$$

$$\text{BranchedAttention}(Q, K, V) = \sum_{i=1}^{M} \alpha_i \text{FFN}_i(\overline{\text{head}}_i). \tag{7}$$

where $M$ denotes the total number of branches, $\kappa_i, \alpha_i \in \mathbb{R}^+$ are learned parameters and $W^{O_i} \in \mathbb{R}^{d_v \times d_{\text{model}}}$. The FFN functions above are identical in form to Equation (4) but since there are $M$ of

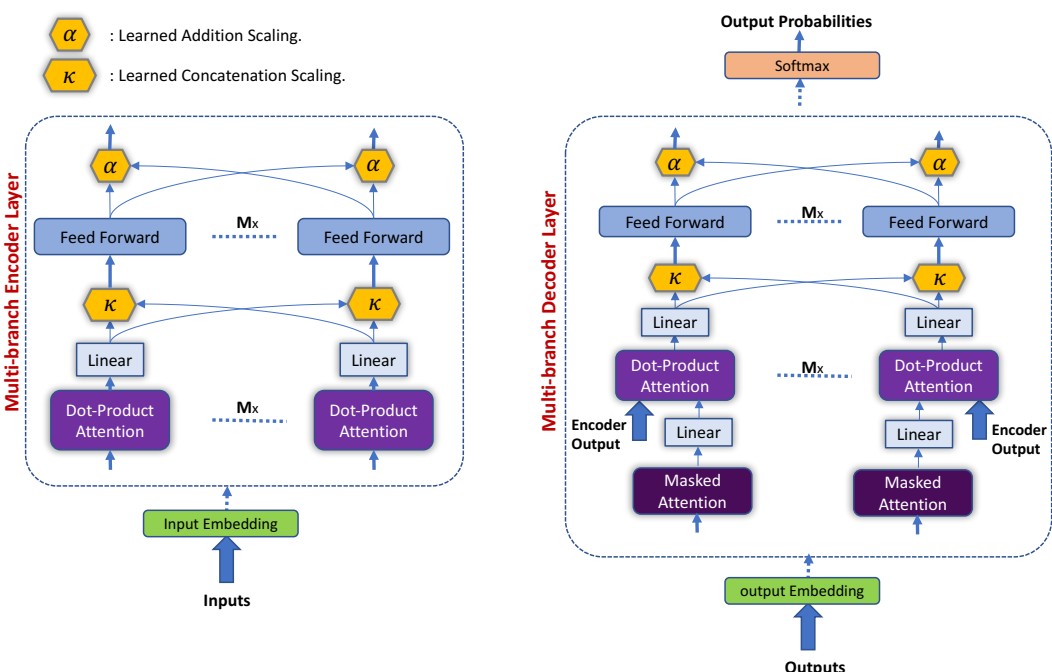

Figure 1: Our proposed network architecture.

them, they have commensurately reduced dimensionality to ensure that no additional parameters are added to the network. Further, we require that $\sum \kappa_i = 1$ and $\sum \alpha_i = 1$ so that Equation (7) is a weighted sum of the individual branch attention values.

We now briefly contrast the modified architecture with the base Transformer model. In the same notation as (5)–(7), the attention layer in the base model can be described as:

$$\text{head}_i = \text{Attention}(QW_i^Q, KW_i^K, VW_i^V), \tag{8}$$

$$\overline{\text{head}}_i = \text{head}_i W^{O_i}, \tag{9}$$

$$\text{BranchedAttention}(Q, K, V) = \text{FFN}(\sum_{i=1}^{M} \overline{\text{head}}_i). \tag{10}$$

Instead of aggregating the contributions from the different heads through $W^O$ right away and using a feed-forward sub-layer, we retain $\overline{\text{head}}_i$ for each of the $M$ heads, learn to amplify or diminish their contribution, use a feed-forward sub-layer and then aggregate them, again in a learned fashion. In the equations above, $\kappa$ can be interpreted as a learned *concatenation* weight and $\alpha$ as the learned *addition* weight. Indeed, $\kappa$ scales the contribution of the various branches before $\alpha$ is used to sum them in a weighted fashion. We ensure that the simplex constraint is respected during each training step by projection. Finally, note that our modification does not add depth (i.e., through the FFN sub-layers) to any of the attention head transformation since the feed-forward computation is merely split and not stacked.

One interpretation of our proposed architecture is that it replaces the multi-head attention by a multi-branch attention. Rather than concatenating the contributions of the different heads, they are instead treated as branches that a multi-branch network learns to combine. While it is possible that $\alpha$ and $\kappa$ could be merged into one variable and trained, we found better training outcomes by separating them. It also improves the interpretability of the models gives that $(\alpha, \kappa)$ can be thought of as probability masses on the various branches.

This mechanism adds $\mathcal{O}(M)$ trainable weights. This is an insignificant increase compared to the total number of weights. Indeed, in our experiments, the proposed mechanism added $192$ weights to a model containing $213M$ weights already. Without these additional trainable weights, the proposed

mechanism is identical to the multi-head attention mechanism in the Transformer. The proposed attention mechanism is used in both the encoder and decoder layers and is masked in the decoder layers as in the Transformer network. Similarly, the positional encoding, layer normalization, and residual connections in the encoder-decoder layers are retained. We eliminate these details from Figure 1 for clarity. Instead of using $(\alpha, \kappa)$ learned weights, it is possible to also use a mixture-of-experts normalization via a softmax layer (Shazeer et al., 2017). However, we found this to perform worse than our proposal.

Unlike the Transformer, which weighs all heads equally, the proposed mechanism allows for ascribing importance to different heads. This in turn prioritizes their gradients and eases the optimization process. Further, as is known from multi-branch networks in computer vision (Gastaldi, 2017), such mechanisms tend to cause the branches to learn decorrelated input-output mappings. This reduces co-adaptation and improves generalization. This observation also forms the basis for mixture-of-experts models (Shazeer et al., 2017).

## 4 EXPERIMENTS

### 4.1 TRAINING DETAILS

The weights $\kappa$ and $\alpha$ are initialized randomly, as with the rest of the Transformer weights.

In addition to the layer normalization and residual connections, we use label smoothing with $\epsilon_{ls} = 0.1$, attention dropout, and residual dropout with probability $P_{drop} = 0.1$. Attention dropout randomly drops out elements (Srivastava et al., 2014) from the softmax in (1).

As in Vaswani et al. (2017), we used the Adam optimizer (Kingma & Ba, 2014) with $(\beta_1, \beta_2) = (0.9, 0.98)$ and $\epsilon = 10^{-9}$. We also use the learning rate warm-up strategy for Adam wherein the learning rate $lr$ takes on the form:

$$lr = d_{\text{model}}^{-0.5} \cdot \min(\text{iterations}^{-0.5}, \text{iterations} \cdot 4000^{-1.5}),$$

for the all parameters except $(\alpha, \kappa)$ and

$$lr = (d_{\text{model}}/\text{N})^{-0.5} \cdot \min(\text{iterations}^{-0.5}, \text{iterations} \cdot 400^{-1.5}),$$

for $(\alpha, \kappa)$.

This corresponds to the warm-up strategy used for the original Transformer network except that we use a larger peak learning rate for $(\alpha, \kappa)$ to compensate for their bounds. Further, we found that freezing the weights $(\kappa, \alpha)$ in the last $10K$ iterations aids convergence. During this time, we continue training the rest of the network. We hypothesize that this freezing process helps stabilize the rest of the network weights given the weighting scheme.

We note that the number of iterations required for convergence to the final score is substantially reduced for the Weighted Transformer. We found that Weighted Transformer converges 15–40% faster as measured by the total number of iterations to achieve optimal performance. We train the baseline model for 100K steps for the smaller variant and 300K for the larger. We train the Weighted Transformer for the respective variants for 60K and 250K iterations. We found that the objective did not significantly improve by running it for longer. Further, we do not use any averaging strategies employed in Vaswani et al. (2017) and simply return the final model for testing purposes.

In order to reduce the computational load associated with padding, sentences were batched such that they were approximately of the same length. All sentences were encoded using byte-pair encoding (Sennrich et al., 2015) and shared a common vocabulary. Weights for word embeddings were tied to corresponding entries in the final softmax layer (Inan et al., 2016; Press & Wolf, 2016). We trained all our networks on NVIDIA K80 GPUs with a batch containing roughly 25,000 source and target tokens.

### 4.2 RESULTS ON BENCHMARK DATA SETS

We benchmark our proposed architecture on the WMT 2014 English-to-German and English-to-French tasks. The WMT 2014 English-to-German data set contains 4.5M sentence pairs. The English-to-French contains 36M sentence pairs.

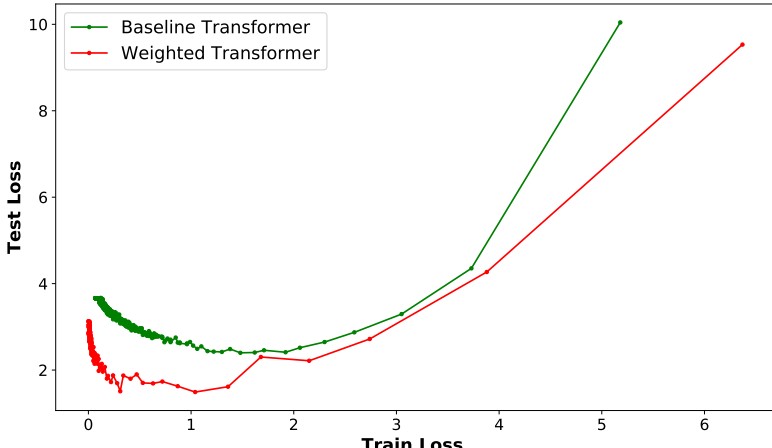

Figure 2: Testing v/s Training Loss for the newstest2013 English-to-German task. The Weighted Transformer has lower testing loss compared to the baseline Transformer for the same training loss, suggesting a regularizing effect.

| Model | EN-DE BLEU | EN-FR BLEU |
|---|---|---|
| Transformer (small) (Vaswani et al., 2017) | 27.3 | 38.1 |
| **Weighted Transformer** (small) | **28.4** | **38.9** |
| Transformer (large) (Vaswani et al., 2017) | 28.4 | 41.0 |
| **Weighted Transformer** (large) | **28.9** | **41.4** |
| ByteNet (Kalchbrenner et al., 2016) | 23.7 | - |
| Deep-Att+PosUnk (Zhou et al., 2016) | - | 39.2 |
| GNMT+RL (Wu et al., 2016) | 24.6 | 39.9 |
| ConvS2S (Gehring et al., 2017) | 25.2 | 40.5 |
| MoE (Shazeer et al., 2017) | 26.0 | 40.6 |

Table 1: Experimental results on the WMT 2014 English-to-German (EN-DE) and English-to-French (EN-FR) translation tasks. Our proposed model outperforms the state-of-the-art models including the Transformer (Vaswani et al., 2017). The small model corresponds to configuration (A) in Table 2 while large corresponds to configuration (B).

Results of our experiments are summarized in Table 1. The Weighted Transformer achieves a 1.1 BLEU score improvement over the state-of-the-art on the English-to-German task for the smaller network and 0.5 BLEU improvement for the larger network. In the case of the larger English-to-French task, we note a 0.8 BLEU improvement for the smaller model and a 0.4 improvement for the larger model. Also, note that the performance of the smaller model for Weighted Transformer is close to that of the larger baseline model, especially for the English-to-German task. This suggests that the Weighted Transformer better utilizes available model capacity since it needs only 30% of the parameters as the baseline transformer for matching its performance. Our relative improvements do not hinge on using the BLEU scores for comparison; experiments with the GLEU score proposed in Wu et al. (2016) also yielded similar improvements.

Finally, we comment on the regularizing effect of the Weighted Transformer. Given the improved results, a natural question is whether the results stem from improved regularization of the model. To investigate this, we report the testing loss of the Weighted Transformer and the baseline Transformer against the training loss in Figure 2. Models which have a regularizing effect tend to have lower testing losses for the same training loss. We see this effect in our experiments suggesting that the proposed architecture may have better regularizing properties. This is not unexpected given similar outcomes for other branching-based strategies such as Shake-Shake Gastaldi (2017) and mixture-of-experts Shazeer et al. (2017).

| Model | Settings | | | | | | | BLEU | params |
|---|---|---|---|---|---|---|---|---|---|
| | $N$ | $d_{model}$ | $d_{ff}$ | $h$ | $M$ | $P_{drop}$ | train steps | | $\times 10^6$ |
| Transformer (C) | 2 | 512 | 2048 | 8 | NA | 0.1 | 100K | 23.7 | 36 |
| **Weighted Transformer** (C) | 2 | 512 | 2048 | 8 | 8 | 0.1 | 60K | **24.8** | 36 |
| Transformer | 4 | 512 | 2048 | 8 | NA | 0.1 | 100K | 25.3 | 50 |
| **Weighted Transformer** | 4 | 512 | 2048 | 8 | 8 | 0.1 | 60K | **26.2** | 50 |
| Transformer (A) | 6 | 512 | 2048 | 8 | NA | 0.1 | 100K | 25.8 | 65 |
| **Weighted Transformer** (A) | 6 | 512 | 2048 | 8 | 8 | 0.1 | 60K | **26.5** | 65 |
| Transformer | 8 | 512 | 2048 | 8 | NA | 0.1 | 100K | 25.5 | 80 |
| **Weighted Transformer** | 8 | 512 | 2048 | 8 | 8 | 0.3 | 60K | **25.6** | 80 |
| Transformer (B) | 6 | 1024 | 4096 | 16 | NA | 0.3 | 300K | 26.4 | 213 |
| **Weighted Transformer** (B) | 6 | 1024 | 4096 | 16 | 16 | 0.3 | 250K | **27.2** | 213 |

Table 2: Experimental comparison between different variants of the Transformer (Vaswani et al., 2017) architecture and our proposed Weighted Transformer. Reported BLEU scores are evaluated on the English-to-German translation development set, newstest2013.

| Model | BLEU |
|---|---|
| Weighted Transformer | 24.8 |
| Train $\kappa$, $\alpha$ fixed to 1 | 24.5 |
| Train $\alpha$, $\kappa$ fixed to 1 | 23.9 |
| $\alpha, \kappa$ both fixed to 1 | 23.6 |
| Without the simplex constraints | 24.5 |

Table 3: Model ablations of Weighted Transformer on the newstest2013 English-to-German task for configuration (C). This shows that the learning both $(\alpha, \kappa)$ and retaining the simplex constraints are critical for its performance.

## 4.3 SENSITIVITY ANALYSIS

In Table 2, we report sensitivity results on the newstest2013 English-to-German task. Specifically, we vary the number of layers in the encoder/decoder and compare the performance of the Weighted Transformer and the Transformer baseline. Using the same notation as used in the original Transformer network, we label our configurations as (A), (B) and (C) with (C) being the smallest. The results clearly demonstrate the benefit of the branched attention; for every experiment, the Weighted Transformer outperforms the baseline transformer, in some cases by up to 1.3 BLEU points. As in the case of the baseline Transformer, increasing the number of layers does not necessarily improve performance; a modest improvement is seen when the number of layers $N$ is increased from 2 to 4 and 4 to 6 but the performance degrades when $N$ is increased to 8. Increasing the number of heads from 8 to 16 in configuration (A) yielded an even better BLEU score. However, preliminary experiments with $h = 16$ and $h = 32$, like in the case with $N$, degrade the performance of the model.

In Figure 3, we present the behavior of the weights $(\alpha, \kappa)$ for the second encoder layer of the configuration (C) for the English-to-German newstest2013 task. The figure shows that, in terms of relative weights, the network does prioritize some branches more than others; circumstantially by as much as $2\times$. Further, the relative ordering of the branches changes over time suggesting that the network is not purely exploitative. A purely exploitative network, which would learn to exploit a subset of the branches at the expense of the rest, would not be preferred since it would effectively reduce the number of available parameters and limit the representational power. Similar results are seen for other layers, including the decoder layers; we omit them for brevity.

Finally, we present an ablation study to highlight the ingredients of our proposal that assisted the improved BLEU score in Table 3. The results show that having both $\alpha$ and $\kappa$ as learned, in conjunction with the simplex constraint, was necessary for improved performance.

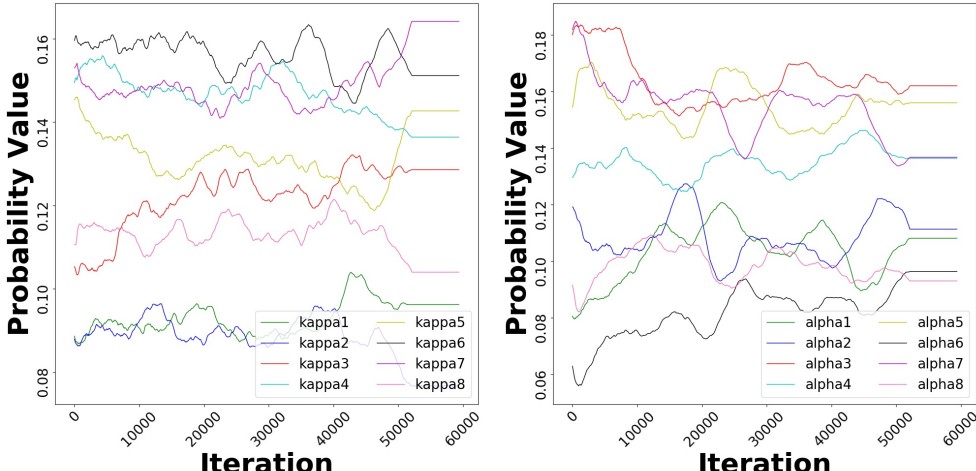

Figure 3: Convergence of the $(\alpha, \kappa)$ weights for the second encoder layer of Configuration (C) for the English-to-German newstest2013 task. We smoothen the curves using a mean filter. This shows that the network does prioritize some branches more than others and that the architecture does not exploit a subset of the branches while ignoring others.

| Weights $(\alpha, \kappa)$ | BLEU |
|---|---|
| Learned | 24.8 |
| Random | 21.1 |
| Uniform | 23.4 |

Table 4: Performance of the architecture with random and uniform normalization weights on the newstest2013 English-to-German task for configuration (C). This shows that the learned $(\alpha, \kappa)$ weights of the Weighted Transformer are crucial to its performance.

### 4.4 RANDOMIZATION BASELINE

The proposed modification can also be interpreted as a form of Shake-Shake regularization proposed in Gastaldi (2017). In this regularization strategy, random weights are sampled during forward and backward passes for weighing the various branches in a multi-branch network. During test time, they are weighed equally. In our strategy, the weights are *learned* instead of being sampled randomly. Consequently, no changes to the model are required during test time.

In order to better understand whether the network benefits from the learned weights or if, at test time, random or uniform weights suffice, we propose the following experiment: the weights for the Weighted Transformer, including $(\alpha, \kappa)$ are trained as before, but, during test time, we replace them with (a) randomly sampled weights, and (b) $1/M$ where $M$ is the number of incoming branches. In Table 4, we report experimental results on the configuration (C) of the Weighted Transformer on the English-to-German newstest2013 data set (see Table 2 for details regarding the configuration). It is evident that random or uniform weights cannot replace the learned weights during test time. Preliminary experiments suggest that a Shake-Shake-like strategy where the weights are sampled randomly during training also leads to inferior performance.

### 4.5 GATING

In order to analyze whether a hard (discrete) choice through gating will outperform our normalization strategy, we experimented with using gates instead of the proposed concatenation-addition strategy. Specifically, we replaced the summation in Equation (7) by a gating structure that sums up the contributions of the top $k$ branches with the highest probabilities. This is similar to the sparsely-gated mixture of experts model in Shazeer et al. (2017). Despite significant hyper-parameter tuning of $k$ and $M$, we found that this strategy performs worse than our proposed mechanism by a large

margin. We hypothesize that this is due to the fact that the number of branches is low, typically less than 16. Hence, sparsely-gated models lose representational power due to reduced capacity in the model. We plan to investigate the setup with a large number of branches and sparse gates in future work.

## 5 CONCLUSIONS

We present the Weighted Transformer that trains faster and achieves better performance than the original Transformer network. The proposed architecture replaces the multi-head attention in the Transformer network by a multiple self-attention branches whose contributions are learned as a part of the training process. We report numerical results on the WMT 2014 English-to-German and English-to-French tasks and show that the Weighted Transformer improves the state-of-the-art BLEU scores by $0.5$ and $0.4$ points respectively. Further, our proposed architecture trains $15 - 40\%$ faster than the baseline Transformer. Finally, we present evidence suggesting the regularizing effect of the proposal and emphasize that the relative improvement in BLEU score is observed across various hyper-parameter settings for both small and large models.

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
