# OpenReview forum: "Weighted Transformer Network for Machine Translation"
_ICLR.cc/2018/Conference — Reject_

### Official Review · AnonReviewer1 · 2017-11-27
**Small tweaks ot the multi-head attention from "Attention is all you need"**

**Rating:** 4
**Confidence:** 4

**Review:**

The paper presentes a small extension to the Neural Transformer model of Vaswani et al 2017:
the multi-head attention computation (eq. 2,3):
head_i = Attention_i(Q,K,W)
MultiHead = Concat_i(head_i) * W = \sum_i head_i * W_i

is replaced with the so-called BranchedAttention (eq. 5,6,7,4):
head_i = Attention_i(Q,K,W)       // same as in the base model
BranchedAttention = \sum_i \alpha_i max(0, head_i * W_i * kappa_i * W^1 + b^1) W^2 + b^2

The main difference is that the results of application of each attention head is post-processed with a 2-layer ReLU network before being summed into the aggregated attention vector.

My main problem with the paper is understanding what really is implemented: the paper states that with alpha_i=1 and kappa_i=1 the two attention mechanism are equivalent. The equations, however, tell a different story: the original MultiHead attention quickly aggregates all attention heads, while the proposed BranchedAttention adds another processing step, effectively adding depth to the model.

Since the BranchedAttention is the key novelty of the paper, I am confused by this contradiction and treat it as a fatal flaw of this paper (I am willing to revise my score if the authors explain the equations) - the proposed attention either adds a small amount of parameters (the alphas and kappas) that can be absorbed by the other weights of the network, and the added alphas and kappas are easier/faster to optimize, as the authors state in the text, or the BranchedAttention works as shown in the equations, and effectively adds depth to the network by processing each attention's result with a small MLP before combining multiple attention heads. This has to be clarified before the paper is published.

The experiment show that the proposed change speeds convergence and improves the results by about 1 BLEU point. However, this requires a different learning rate schedule for the introduced parameters and some non-standard tricks, such as freezing the alphas and kappas during the end of the training.

I also have a questions about the presented results:
1) The numbers for the original transformer match the ones in Vaswani et al 2017, am I correct to assume that the authors did not rerun the tensor2tensor code and simply copied them from the paper?
2) Is all of the experimental setup the same as in Vaswani et al 2017? Are the results obtained using their tensor2tensor implementation, or are some hyperparameters different?

Detailed review:
Quality:
The equations and text in the paper contradict each other.

Clarity:
The language is clear, but the main contribution could be better explained.

Originality:
The proposed change is a small extension to the Neural Transformer model.

Significance:
Rather small, the proposed addition adds little modeling power to the network and its advantage may vanish with more data/different learning rate schedule.

Pros and cons:
+ the proposed approach is a simple way to improve the performance of multihead attentional models.
- it is not clear from the paper how the proposed extension works: does it regularize the model or dies it increase its capacity?

---

> ### Author Response · Authors · 2017-12-16
> **Reply to AnonReviewer1**
>
> We thank you for your insightful review. We clarify the questions below.
>
> “the paper states that with alpha_i=1 and kappa_i=1 the two attention mechanism are equivalent. The equations, however, tell a different story: the original MultiHead attention quickly aggregates all attention heads, while the proposed BranchedAttention adds another processing step, effectively adding depth to the model.”
> We found that the equivalence statement is indeed untrue given the nonlinearity of the FFN layer and have removed it from our updated manuscript.. However, we emphasize that our modification does not add depth to the network. Each layer of the original Transformer network consists of three operations (ignoring the residual connections and layer normalization): computation of the attention head, concatenation of the heads and finally the FFN layer. Our proposed architecture is a replacement for the entire layer (consisting of all three operations) and not just the multi-head attention. In doing so, we also change the order of operations; unlike the Transformer which concatenates the heads and then projects them through a single FFN layer, we first use $M$ FFN layers, one for each of the heads, and then combine them. The FFN layers, in this case, will be of commensurately reduced sizes (the original dimension divided by the number of the branches). Owing to this, the original and proposed Transformer networks have the same depth and, barring the scalar (alpha, kappa) values, exactly the same number of trainable parameters. We apologize for the lack of clarity and have fixed our notation and added an explanation in our updated manuscript. Specifically, we have expressed and contrasted the original Transformer network in the notation of (5)--(7).
>
> “The numbers for the original transformer match the ones in Vaswani et al 2017, am I correct to assume that the authors did not rerun the tensor2tensor code and simply copied them from the paper?”
> That is true. Having said that, we have experimented with the baseline on our (home-grown) implementation and found very similar metrics for the original Transformer as those reported by the authors.
>
> “Is all of the experimental setup the same as in Vaswani et al 2017? Are the results obtained using their tensor2tensor implementation, or are some hyperparameters different?”
> It is the same as Vaswani et. al. We used the tensor2tensor code as foundation for our modified architecture and, other than the proposed branching mechanism, all else stayed the same.

---

### Official Review · AnonReviewer2 · 2017-11-27
**Improved results but unsure about details**

**Rating:** 6
**Confidence:** 5

**Review:**

TL;DR of paper: they modify the Transformer architecture of Vaswani et al. (2017) to used branched attention with learned weights instead of concatenated attention, and achieve improved results on machine translation.

Using branches instead of a single path has become a hot architecture choice recently, and this paper applies the branching concept to multi-head attention. Weirdly, they propose using two different sets of weights for each branch: (a) kappa, which premultiplies the head before fully connected layers, and (b) alpha, which are the weights of the sum of the heads after the fully connected layers. Both weights have simplex constraints. A couple of questions about this:

* What is the performance of only using kappa? Only alpha? Neither? What happens if I train only of them?
* What happens if you remove the simplex constraints (i.e., don't have to sum to one, or can be negative)?
* Why learn a global set of weights for the branch combiners? What happens if the weights are predicted for each input example? This is the MoE experiment, but where k = M (i.e., no discrete choices made).
* Are the FFN layer parameters shared across the different heads?
* At the top of page 4, it is said "all bounds are respected during each training step by projection". What does this mean? Is projected gradient descent used, or is a softmax used? If the former, why not use a softmax?
* In Figure 3, it looks like the kappa and alpha values are still changing significantly before they are frozen. What happens if you let them train longer? On the same note, the claim is that Transformer takes longer to train. What is the performance of Transformer if using the same number of steps as the weighted Transformer?
* What are the Transformer variants A, B, and C?

While the results are an improvement over the baseline Transformer, my main concern with this paper is that the improved results are because of extensive hyperparameter tuning. Design choices like having a separate learning rate schedule for the alpha and kappa parameters, and needing to freeze them at the end of training stoke this concern. I'm happy to change my score if the authors can provide empirical evidence for each design choice

---

> ### Author Response · Authors · 2017-12-16
> **Reply to AnonReviewer2**
>
> We thank you for your insightful review. We answer your questions below.
>
> “What is the performance of only using kappa? Only alpha? Neither? What happens if I train only of them? What happens if you remove the simplex constraints (i.e., don't have to sum to one, or can be negative)?”
> We experimented with these changes and found that they lead to inferior performance. Here is our summary:
> +-----------------------------------------+---------------------------------------+
> | Model                                          | Performance on Config (C) |
> +------------------------------------------+--------------------------------------+
> | Weighted Transformer             |            24.8                              |
> +------------------------------------------+--------------------------------------+
> | Only kappa, alpha=1                 |            24.5                              |
> +------------------------------------------+--------------------------------------+
> | Only alpha, kappa=1                 |            23.9                              |
> +------------------------------------------+--------------------------------------+
> | alpha=1, kappa=1                      |            23.6                              |
> +------------------------------------------+--------------------------------------+
> | No simplex constraints             |            24.5                              |
> +------------------------------------------+--------------------------------------+
> | Without freezing of weights    |            24.7                              |
> +------------------------------------------+--------------------------------------+
> We expect similar results on other configurations given our previous experiments and have added this Table to our paper.
>
> “Why learn a global set of weights for the branch combiners...”
> We learn the (alpha, kappa) for each layer individually and not for the entire network.
>
> “Are the FFN layer parameters shared across the different heads?”
> No, each head has a separate set of parameters as in the case of the original Transformer network. However, our FFN weight matrices are commensurately smaller in size.
>
> “At the top of page 4, it is said "all bounds are respected during each training step by projection...”
> We employ projected gradient descent;  we experimented with using a softmax weighting but found it be slower, more noisy in its updates, performed worse by a significant margin over a projected score.
>
> “In Figure 3, it looks like the kappa and alpha values are still changing significantly before they are frozen...”
> Training the original Transformer network for fewer iterations led to inferior performance in all cases. On the other hand, training our Weighted Transformer beyond the described threshold did not help our BLEU scores. Both observations were also buttressed by our training curves.
>
> “What are the Transformer variants A, B, and C?”
> They are Weighted Transformers of three different configurations (which we describe in Table 2). We have expanded upon this notation in the text in our updated manuscript.

---

### Official Review · AnonReviewer3 · 2017-11-28
**review of "Weighted transformer network for machine translation"**

**Rating:** 9
**Confidence:** 4

**Review:**

This paper describes an extension to the recently introduced Transformer networks which shows better convergence properties and also improves results on standard machine translation benchmarks.

This is a great paper -- it introduces a relatively simple extension of Transformer networks which only adds very few parameters and speeds up convergence and achieves better results. It would have been good to also add a motivation for doing this (for example, this idea can be interpreted as having a variable number of attention heads which can be blended in and out with a single learned parameter, hence making it easier to use the parameters where they are needed). Also, it would be interesting to see how important the concatenation weight and the addition weight are relative to each other -- do you possibly get the same results even without the concatenation weight?

A suggested improvement: Please check the references in the introduction and see if you can find earlier ones -- for example, language modeling with RNNs has been done for a very long time, not just since 2017 which are the ones you list; similar for speech recognition etc. (which probably has been done since 1993!).

Addition to the original review: Your added additional results table clarifies a lot, thank you. As for general references for RNNs, I am not sure Hochreiter & Schmidhuber 1997 is a good reference as this only points to a particular type of RNN that is used today a lot. For speech recognition there are many better citations as well, check the conference proceedings from ICASSP for papers from Microsoft, Google, IBM, which are the leaders in speech recognition technology. However, I know citations can be difficult to get right for everybody, just try to do your best.

---

> ### Author Response · Authors · 2017-12-16
> **Reply to AnonReviewer3**
>
> We thank you for your review and your assessment. We answer your questions below.
>
> “...it would be interesting to see how important the concatenation weight and the addition weight ...”
> We experimented with these and other changes and found that they lead to inferior performance. Here is our summary:
>
> +-----------------------------------------+---------------------------------------+
> | Model                                          | Performance on Config (C) |
> +------------------------------------------+--------------------------------------+
> | Weighted Transformer             |            24.8                              |
> +------------------------------------------+--------------------------------------+
> | Only kappa, alpha=1                 |            24.5                              |
> +------------------------------------------+--------------------------------------+
> | Only alpha, kappa=1                 |            23.9                              |
> +------------------------------------------+--------------------------------------+
> | alpha=1, kappa=1                      |            23.6                              |
> +------------------------------------------+--------------------------------------+
> | No simplex constraints             |            24.5                              |
> +------------------------------------------+--------------------------------------+
> | Without freezing of weights    |            24.7                              |
> +------------------------------------------+--------------------------------------+
> We expect similar results on other configurations given our previous experiments and have added this Table to our paper.
>
> “...language modeling with RNNs has been done for a very long time, not just since 2017 which are the ones you list; similar for speech recognition etc...”
> We agree and have fixed said references in our updated manuscript.

---

### Public Comment · (anonymous) · 2017-11-20
**Equations for Multi-branch Decoder**

It is not clear to me how the multi-branch decoder layer is constructed. Specific equations for this layer are not present in the paper. It is clear from figure 1 how equations 5-7 are implemented in the encoder and the decoder side (Dot-Product Attention + Linear * \kappa), but it is unclear how exactly the Masked Attention + Linear modules are implemented (compared to the original Transformer model; is it the same or different, and if so how?). Can you please provide the equations for these modules as well?

---

> ### Author Response · Authors · 2017-11-21
> **Multi-Branch Decoder**
>
> The Masked Attention and the Linear modules are implemented as mentioned in the original Transformer paper "Attention Is All You Need" by Vaswani et al. (2017). For the original implementation by Vaswani et al., please visit this github link: https://github.com/tensorflow/tensor2tensor/blob/75b75f2e2281101b9b3637e14ef519afd6a11b68/tensor2tensor/layers/common_attention.py

---

### Public Comment · (anonymous) · 2017-12-01
**Memory and Computational Requirements?**

Based on our understanding of Weighted Transformer, the FFN is applied separately to each of the heads before they are summed up. As a result, our implementation of Weighted Transformer requires M=8 times more memory and time to compute this part. (In practice our implementation of the Weighted Transformer performs 1.5 steps/s vs the standard Transformer’s 2.5 steps/s on a P100).

Is our understanding correct? And if so, should one interpret the claim that the Weighted Transformer “converges 15-40% faster” in terms of *training steps* and not wall clock time?

Furthermore, because the FFN in Branched Attention is applied separately to each branch and the outputs are then summed (eqn 7), i.e. sum_i alpha_i (FFN(\bar{head}_i)) as opposed to FFN(sum_i (head_i)) for the standard transformer, we don’t see how  setting kappas and alphas to 1 reduces to the multi-head attention of the Transformer (due to the non-linearity in the FFN). Can you please clarify?

---

> ### Author Response · Authors · 2017-12-30
> **Reply - Memory and Computational Requirements**
>
> Focusing on the FFN layers, while they are separately applied and added up, each of them is of commensurately smaller size and hence does not add to the number of parameters. Hence, the memory and compute requirements are comparable to the baseline Transformer network.

---

### Decision · Program_Chairs · 2018-01-29
**ICLR 2018 Conference Acceptance Decision**

**Decision:**

Reject

**Comment:**

The paper proposes a modification to the Transformer network, which mostly consists in changing how the attention heads are combined. The contribution is incremental, and its novelty is limited. The results demonstrate an improvement over the baseline at the cost of a more complicated training procedure with more hyper-parameters, and it is possible that with similar tuning the baseline performance could be improved in a similar way.